# Enhanced Collagen Prolyl 4-Hydroxylase Activity and Expression Promote Cancer Progression via Both Canonical and Non-Canonical Mechanisms

**DOI:** 10.3390/ijms26199371

**Published:** 2025-09-25

**Authors:** Dalton Hironaka, Gaofeng Xiong

**Affiliations:** 1Molecular, Cellular, and Developmental Biology Program, Department of Veterinary Biosciences, The Ohio State University, Columbus, OH 43210, USA; hironaka.1@buckeyemail.osu.edu; 2Department of Veterinary Biosciences, Comprehensive Cancer Center, The Ohio State University, Columbus, OH 43210, USA

**Keywords:** collagen, prolyl 4-hydroxylase, tumor microenvironment

## Abstract

Collagens make up the main components of the extracellular matrix (ECM), and, in cancer, are often aberrantly secreted by both tumor cells and stromal cells in the tumor microenvironment (TME). Collagen prolyl 4-hydroxylase (C-P4H), an enzyme that hydroxylates proline into 4-hydroxyproline at the Y position of the collagen -X-Y-Gly- triplet motif, is essential for the stability of the mature collagen trimer and collagen secretion. In this review, we summarize the research on the structure and function of C-P4H, the regulation of C-P4H enzyme activity, and the role of overexpression of its α-subunit, P4HA1, in promoting cancer progression as well as its potential as a prognostic marker and therapeutic target. Overexpression of P4HA1 is displayed in almost all solid cancers, including breast, colorectal, and lung cancer, and is associated with cancer progression, worse response to therapy, and poorer patient survival. Characterization of P4HA1 overexpression has demonstrated links to key hallmarks of cancer, not only in the canonical collagen deposition role, but also in non-canonical functions, such as cell stemness, hypoxic response, glucose metabolism, angiogenesis, and modulation of tumor-infiltrating lymphocytes (TILs) in the tumor microenvironment. P4HA1 is thus an attractive target for developing novel targeted therapies to improve treatment response in many cancer types.

## 1. Structure of Collagen Prolyl 4-Hydroxylase

Collagen prolyl 4-hydroxylase (C-P4H) is a member of the α-ketoglutarate-dependent (α-KG-dependent) dioxygenase family. It is a tetrameric, α_2_β_2_ enzyme in which the two α subunits are made up of one of three types, coded by P4HA1, P4HA2, and P4HA3, and the two β subunits consist of a protein disulfide isomerase (PDI), coded by P4HB [1,2]. Each α subunit does not mix with the other α subunits when forming the C-P4H tetramer [2,3] and has different expression patterns; P4HA1 is predominant in most cells, while P4HA2 is the main type in chondrocytes and endothelial cells [4], and P4HA3 is expressed in many cell types but at lower levels than P4HA1 and P4HA2 [2]. P4HA1 and P4HA2 are known to have at least two mutually exclusive alternative mRNA transcripts, while P4HA3 does not [5]. For P4HA1, both transcripts are similarly expressed in all tissues, suggesting similar functions despite having relatively low sequence and amino acid identity [1,2]. The α subunit is translated on ribosomes along the rough endoplasmic reticulum (ER), after which they then bind to the β subunit, which is constitutively present in the ER for other chaperone functions (Figure 1). P4HB has a KDEL retention sequence to keep C-P4H in the ER lumen [6]. P4HA1 and P4HB are also able to localize to the mitochondria, where the C-P4H holoenzyme is active, although the substrate(s) and mechanism of localization have not yet been identified [7]. The β subunit P4HB is a PDI, coded by the same gene as PDIA1, and has functions other than its involvement in the C-P4H tetramer, such as acting as a chaperone protein [8,9]. Due to its other functions, P4HB is endogenously synthesized at much higher levels than the three α subunits (Figure 1), so the amount of active C-P4H holoenzyme is regulated by expression of the P4HA genes [9]. However, P4HB does not appear to have detectable function when associated with P4HA, and its catalytic sites can be functionally deactivated without affecting C-P4H activity [10,11]. In C-P4H, P4HB is required for the solubility of the α subunit, which otherwise aggregates in solution [10,12].

Crystal structures for both the C-P4H tetramer and the individual subunits have been difficult to obtain due to the low solubility of the α subunit, and, currently, there is an incomplete understanding of the catalytic site and how the α and β subunits assemble into a heteroteramer. The P4HA isomers have four domains: the N-terminal dimerization domain, peptide-substrate-binding (PSB) domain, a linker region, and the C-terminal catalytic (CAT) domain [13,14,15]. The structures of the dimerization and PSB domains have been determined [15,16], showing dimerization via a coiled-coil helical dimerization motif [15] and substrate binding in the PSB domain via a groove containing conserved tyrosine residues [17]. However, the CAT domain crystal structure remains unsolved due to its high insolubility. A crystal structure of an algal homolog for P4HA has been solved, which shows a similar active site to those found in other α-KG-dependent dioxygenases, consisting of antiparallel β-strands forming a double-stranded β-helix (DSBH) fold with a space between the two sheets for α-KG to bind as a cofactor [18]. P4HB has four domains, named **a**, **a′**, **b**, and **b′**, each with a thioredoxin fold that has different conformations depending on its redox state [19] and crystal structures, which have been determined for both their oxidized and reduced conformations [20]. These crystal structures show the four domains of P4HB organized in a horseshoe shape, containing one active site at each end in the **a** and **a′** domains, which are brought closer together when the thioredoxin folds are reduced and further apart when they are oxidized [20]. However, the relevance of this function for C-P4H assembly and activity is unclear, as the **b′** and **a′** domains alone are sufficient for heterotetramer function [21]. Recently, a small-angle X-ray scattering structure of the full heterotetramer [13] and a crystal structure for the P4HA2 CAT domain complexed with P4HB [22] have been solved. These structures show that C-P4H is organized in a βααβ manner, with the β subunits at each end and each αβ dimer linked in the middle by the α subunits’ dimerization domains. The CAT domain of the α subunit interacts with the β subunit via its **a**, **b′**, and **a′** domains, involving a disulfide bridge with the **a′** domain and hydrophobic interactions with the **b′** domain, facing the CAT domain’s catalytic site to the bulk solvent, where it can interact with substrate [22]. Possible interactions between the CAT domain and the dimerization domains have been predicted but remain to be fully understood.

## 2. Function of C-P4H

C-P4H is a member of the Fe^2+^ and α-KG-dependent dioxygenase family, requiring molecular oxygen and α-KG to be consumed stoichiometrically to hydroxylate proline residues, releasing CO_2_ and succinate as byproducts [23]. The Fe^2+^ ion is coordinated by water from the solvent and three side chains of the P4HA CAT domain in a His-X-Asp motif, which have not been fully identified in P4HA1 but correspond to His430, His501, and Asp432 in P4HA2 [22]. The exact mechanism of the reaction catalyzed by C-P4H has not been described; however, based on studies of homologs and other α-KG-dependent dioxygenases, it is thought that α-KG binds first, followed by proline on the procollagen and O_2_, displacing the coordinated water molecules. The reaction then proceeds through two steps: first, decarboxylating α-KG to succinate, releasing CO_2_, and then using the resulting reactive ferryl ion to hydroxylate proline at the C-4 position [24,25]. Ascorbate is also a co-factor but is not consumed stoichiometrically and is instead involved in regenerating the Fe^2+^ ion [23]. Uncoupled reactions can occur via an unknown mechanism in which proline is not hydroxylated, leaving an unreactive Fe(III) bound to OH and succinate, requiring ascorbate to reactivate the enzyme by acting as an electron donor [24]. Some cycles of catalysis can proceed in the absence of ascorbate, but even when substrate is present at saturating concentrations, uncoupled decarboxylation may occur [1].

The primary function of C-P4H is to hydroxylate proline residues at carbon 4 (C-4) in the Y-position of an -X-Y-Gly- triplet sequence motif that is found in procollagen peptides, which is essential for the proper formation of the tertiary collagen triple helix [23]. A recent study has also demonstrated an alternate binding mechanism for C-P4H in which the Y-position amino acid is not proline, and, instead, a proline at the X-position is hydroxylated [26], although the function of this alternate binding and its biological relevance have not yet been described. Hydroxyproline has an abundance of about 4% in animal proteins, most of which is found in collagen, making it highly abundant relative even to many unmodified residues [25]. Not much is known about P4HA3 binding and activity, but it is known that the tetramers formed by P4HA1 and P4HA2, respectively, appear to have different substrate binding affinities. Loss of P4HA1 leads to a decrease in hydroxylation primarily at collagen triplets where the X-position amino acid has a positively charged or uncharged polar side chain, while loss of P4HA2 leads to a decrease in hydroxylation of those with a negatively charged side chain [27]. This is also supported by data using polypeptides with specific amino acids in the X-position, where there are marked differences between the two isoforms in their ability to bind and hydroxylate those with Asp, Lys, and Asn residues in that position [27]. They also have different affinities for the competitive inhibitor poly(L-proline), a polypeptide that serves as a suitable substrate for many plant prolyl 4-hydroxylases, with their K_i_ values differing by 200–1000 times depending on the length of the polypeptide [3,28]. Substitutions of glutamate and glutamine at residues corresponding to Ile182 and Tyr233, respectively, in the P4HA2 substrate-binding domain compared to P4HA1 likely account for the lack of its ability to bind poly(L-proline) effectively [14]. Crystallization of the P4HA2 structure shows rearrangements of other conserved residues, for example, Tyr196 and Arg223, which may be significant enough to cause differences in contacts in the crystal structure, possibly explaining other differences in substrate binding affinities as described above [17]. P4HA1 and P4HA2 do not show collagen type specificity in vitro, with each holoenzyme being able to hydroxylate different procollagen chains at similar levels [27]. However, P4HA1 and P4HA2 cannot totally substitute for one another, as *P4ha1* knockout in mice is embryonic lethal, while *P4ha2* knockout mice are viable and seem to lack major abnormalities [29,30].

## 3. Regulation of P4HA1 Expression and C-P4H Enzyme Activity in Cancer

A wide variety of transcription factors, signaling molecules, and microRNAs (miRNAs) are known to regulate the expression of P4HA1, both directly and indirectly (Table 1). Expression of the P4HA subunits, especially P4HA1, is known to be induced by hypoxia both in normal states [31,32] and in many cancer types, including oral squamous cell carcinoma [33,34], triple-negative breast cancer [35], and gliomas [36,37], contributing to the ECM remodeling and fibrosis that is seen in cancer. This is mediated largely by hypoxia-inducible factor 1α (HIF-1α), which can directly promote transcription of P4HA1 in hypoxic conditions [38,39,40,41,42,43], as the P4HA1 promoter has been reported to have both multiple hypoxia response elements (HREs) [44] and a HIF ancillary sequence (HAS), which enables HIF binding and multimerization to promote P4HA1 transcription [41]. This produces a feedback loop in which a reduction in α-KG levels by C-P4H prevents HIF-prolyl hydroxylases (PHDs), which are also α-KG-dependent dioxygenases, from hydroxylating HIF-1α, which would otherwise mark it for ubiquitination and proteasomal degradation [35,45]. P4HA1 expression may also be induced by HIF-2α activity [46], although, thus far, there is no evidence of a similar positive feedback loop as with HIF-1α.

Other transcription factors that are known to be able to directly activate P4HA1 expression include signal transducer and activator of transcription 1 (STAT1) [47] and Sp1 transcription factor (SP1) [48]. Thus far, a few transcription factors have also been identified that can repress P4HA1 expression: Non-POU Domain Containing Octamer Binding (NonO) and JUN/FOS heterodimers (Activator Protein 1, AP-1 transcription complex) were identified to negatively regulate P4HA1 in smooth muscle cells [49] and myocardial fibroblasts [50], respectively. P4HA1 expression can be altered by signaling pathways activated by external signaling molecules such as adiponectin, which promotes P4HA1 expression via extracellular signal-regulated kinases 1 and 2 (ERK1/2) signaling [48], and tumor necrosis factor-α (TNF-α), which inhibits its expression through the ASK1-JNK-NonO signaling pathway [49]. The role of activating transcription factor 3 (ATF3) in regulating P4HA1 is controversial. Multiple studies have demonstrated that ATF3 is able to bind the promoter region of P4HA1 during hypoxia. However, in glioblastoma (GBM), this binding recruits a p300/CBP complex to silence P4HA1 transcription [39] while in breast cancer, ATF3 binding instead directly activates P4HA1 transcription [51]. This highlights that while P4HA1 is upregulated in most solid cancer types, the way in which this is accomplished, and which transcription factors are involved, are context-dependent and may depend on cell- and tissue-intrinsic factors.

Other than transcription factors, P4HA1 expression is also controlled at the translational level, via translation factors and miRNAs. A complex composed of HIF-2α, RNA-binding motif protein 4 (RBM4), and eukaryotic translation initiation factor 4E family member 2 (eIF4E2) is able to increase translation of P4HA1 mRNA during hypoxia by increasing its binding affinity for eIF4E2, which regulates translation initiation [52]. Knockdown of basic leucine zipper and W2 domains 2 (BZW2), a basic leucine zipper protein involved in translation initiation that is upregulated in some cancers, reduces expression of P4HA1 in colorectal adenocarcinoma (COAD) [53]. In hypoxia, the RNA-binding protein nucleolin is cleaved to a form that can increase the translational efficiency of P4HA1 mRNA by binding to the 5′- and 3′-untranslated regions (5′/3′-UTR), resulting in the increase in its activity seen in fibrosarcoma cells [54]. Several miRNAs can bind to the 3′-UTR region of the P4HA1 mRNA, promoting its destruction. A few have been described previously, including miR-30e [55], miR-122 [56,57,58], miR-124 [44,59], miR-335-5p [60], and miR-499a-5p [61]. This regulatory mechanism for P4HA1 also seems to be important for mediating overexpression of P4HA1 in cancer, as multiple cancer types show downregulation of P4HA1-suppressing miRNAs at the same time as P4HA1 overexpression [44,55,56,58]. Enhancer of zeste 2 polycomb repressive complex 2 subunit (EZH2), the catalytic subunit of the polycomb repressive complex 2 (PRC2), is an epigenetic modifier that is often responsible for repressing gene transcription. In both COAD [62] and prostate adenocarcinoma (PRAD) [44], knockdown of EZH2 also decreases P4HA1 expression levels, which may be via indirect regulation as EZH2 can suppress transcription of miR-124 [44]. C-terminal binding protein 1 (CTBP1) is a corepressor protein that is also associated with increased P4HA1 expression in prostate cancer and may cooperate with EZH2 to modulate the expression of miR-124 [44]. miRNA expression may also be reduced by long non-coding RNAs, such as silencing of miR-335-5p by LINC01503 [60] and silencing of miR-122-5p by CDKN2B-AS1 [57]. Aberrant downregulation of these miRNAs likely contributes to cancer progression by enabling increased expression of P4HA1.

There are several known and predicted sites for post-translational modifications (PTMs) on the P4HA1 polypeptide, which may impact its stability, substrate binding, and/or activity. These include multiple serine, threonine, and tyrosine phosphorylation sites, ubiquitination sites, and glycosylation sites throughout the N-terminal and CAT domains [63]. The majority of these have been identified by proteomic screens or predicted using in silico methods; however, the effects of N-glycosylation of P4HA1 at N259 have been investigated in mouse and human fibroblast cells [64]. Supplementation of endogenous vitamin C to mouse embryonic fibroblast (MEF) cells increased collagen secretion while simultaneously increasing glycosylation at N259 but not N113. Vitamin C was found to promote activity of a complex composed of STT3 oligosaccharyltransferase complex catalytic subunit B (STT3B) and magnesium transporter 1 (MAGT1), which facilitated glycosylation of P4HA1 [64]. This glycosylation and increase in collagen secretion were found to be due to increased substrate binding by P4HA1 rather than changes to its activity [64], which agrees with the prior literature on P4HA1 showing that N-glycosylation does not impact enzymatic activity [65]. Although the exact function of this N-glycosylation site and how it works in vivo is unknown, this may indicate that there are undiscovered interactions between the PSB and CAT domains, as N259 is in the CAT domain, not the PSB domain. This is an unexpected finding based on the prior literature showing that these PTMs have no effect on enzyme activity [65], so further research is warranted on these glycosylation sites, as well as the other PTMs on P4HA1, to determine how they impact P4HA1 and C-P4H activity.

The reaction catalyzed by C-P4H can be regulated by the amounts of its cofactors, especially α-KG and succinate, due to their usage in a wide variety of metabolic reactions. C-P4H activity can be altered by the addition or depletion of exogenous α-KG [35,66], resulting in changes to collagen secretion. Alternative sources of α-KG also have an effect on C-P4H activity and collagen secretion, such as lactate, which is metabolized into acetyl-CoA and enters the citric acid cycle, increasing the concentration of α-KG as an intermediate [67,68]. The activity of C-P4H is also affected by the concentration of succinate [39], which is produced from α-KG by the hydroxylation reaction. Addition of exogenous succinate is sufficient to decrease C-P4H activity, α-KG utilization, and collagen secretion [35,39]. Components of lactate metabolism are also involved in enhancing C-P4H activity via uptake and processing of extracellular lactate, producing pyruvate, which increases intracellular α-KG via the citric acid cycle. Inhibition of the major transporter of extracellular lactate, solute carrier family 16 member 1 (SLC16A1 or MCT1) in prostate and head and neck cancer reduces intracellular α-KG, resulting in a decrease in collagen deposition associated with decreased C-P4H activity [67,68]. In prostate cancer, there is also a correlation between P4HA1 and MCT1 expression levels in patient sequencing data [67]. Knockdown of lactate dehydrogenase B (LDHB), the beta subunit of the enzyme that catalyzes the conversion of lactate to pyruvate, has similar effects to knockdown of MCT1 in head and neck cancer and also reduces cancer cell stemness in vitro [68]. P4HA1 and the activity of C-P4H are thus intertwined with the broader metabolome of the cell and surrounding cells in the TME.

A wide variety of signaling molecules and other proteins can induce or repress the expression of P4HA1. Many other proteins are implicated in altering P4HA1 expression, with evidence stemming from aberrant expression in solid cancers. In GBM cells, P4HA1 can be protected from lysosomal degradation by polymerase I and transcript release factor (PTRF) by colocalization of both to the lysosome, preventing P4HA1 from degradation, which is associated with worse patient outcome [39]. Turnover of P4HA1 and its stability via proteasome have been studied less extensively; however, there is evidence in P4HA2 of ubiquitin-directed proteasome-dependent degradation via tripartite motif-containing protein 21 (TRIM21) in papillary thyroid cancer [69]. Although P4HA1 and P4HA2 have relatively low sequence identity [2], there are several predicted ubiquitination sites in their sequences [63], suggesting the possibility of similar regulatory mechanisms at the protein level. Choline dehydrogenase (CHDH) levels are upregulated in colorectal cancer (CRC) and may also serve to regulate P4HA1 via epigenetic modifications, as its activity is associated with histone H3 trimethylation, increased P4HA1 expression, and metastasis in vitro [70]. In a similar vein, overexpression of fatty acid binding protein 7 (FABP7) in HER2+ breast cancer results in increased expression of multiple genes promoting a glycolytic and angiogenic phenotype, including P4HA1, particularly during hypoxia [71].

**Table 1 ijms-26-09371-t001:** Regulation of P4HA1.

Transcription Factors	Up-/Down-Regulated	Cancer or Cell Type/Reference(s)
HIF-1α	Up	Triple-negative breast cancer [35]
		Colorectal adenocarcinoma [40,52]
		Glioblastoma [37,39]
		Pancreatic adenocarcinoma [45,72]
		Head and neck squamous cell carcinoma [68]
		Uveal melanoma [38]
		Human embryonic kidney cells [41]
		Human gingivial fibroblasts and human periodontal ligament cells [32]
USP5 and HIF-2α	Up	ER-positive and triple-negative breast cancer [46]
HIF-2α	Up	Colorectal adenocarcinoma [52]
STAT1	Up	Esophageal squamous cell carcinoma [47]
SP1	Up	Human aortic smooth muscle cells [48]
ATF3	Up	Breast cancer [51]
	Down	Glioblastoma [39]
NonO	Down	Human aortic smooth muscle cells [49]
AP-1	Down	Human myocardial fibroblasts [50]
Translation Factors and Non-coding RNAs		
BZW2	Up	Colorectal adenocarcinoma [53]
RBM4	Up	Colorectal adenocarcinoma [52]
EIF4E2	Up	Colorectal adenocarcinoma [52]
Nucleolin	Up	Fibrosarcoma [54]
		Thyroid cancer [57]
miR-30e	Down	Hepatocellular carcinoma [55]
miR-122	Down	Ovarian cancer [56,57,58,73]
		Thyroid cancer [57]
miR-124	Down	Prostate adenocarcinoma [44,59]
		Lung adenocarcinoma [59]
miR-335-5p	Down	Pancreatic adenocarcinoma [60]
miR-499a-5p	Down	Head and neck squamous cell carcinoma [61]
Others		
STT3B/MAGT1	Up	Mouse embryonic fibroblasts [64]
SLC16A1/MCT1	Up	Prostate adenocarcinoma [67]
		Head and neck squamous cell carcinoma [68]
LDHB	Up	Head and neck squamous cell carcinoma [68]
PTRF	Up	Glioblastoma [39]
CHDH	Up	Colorectal adenocarcinoma [70]
FABP7	Up	HER2+ breast cancer [71]

## 4. P4HA1 Promotes Cell Proliferation, Invasiveness, Cell Stemness, and Chemoresistance

Collagen deposition has been implicated in the pathogenesis of many solid cancers, involving a variety of processes including metabolic signaling, immune modulation, EMT, and metastatic progression [74,75,76]. As the primary α subunit of C-P4H, P4HA1 has been independently implicated in cancer via both its role in collagen deposition and in relation to other aspects of carcinogenesis. P4HA1 is overexpressed in the majority of solid cancers affecting a wide variety of organ systems, ranging from breast cancer and colon cancer to melanoma and esophageal cancer [77,78]. P4HA1 overexpression is a poor prognostic marker in a variety of cancer types, both independently [78,79] and as parts of functionally linked gene signatures [77,80,81,82]. Overexpression of P4HA1 has been implicated in cancer progression not only through its association with prognosis, but it has also been demonstrated to directly influence cellular behavior, including enhanced proliferation, invasion, and stemness across multiple cancer types. These alterations through cell behavior are mediated through a wide variety of downstream signaling pathways related to hypoxia, glucose metabolism, angiogenesis, and alteration of the ECM (Table 2). In some cancer types, while there is in vitro evidence of an association between P4HA1 and cancer progression, the putative downstream mechanism(s) have not yet been described, for example, lung cancer [59], ovarian cancer (OVCAR) [83], and esophageal cancer [47]. This section will review the evidence that P4HA1 is involved in cancer progression in multiple cancer types, both where the putative downstream pathways have been identified and where they have yet to be determined.

P4HA1 overexpression is consistently associated with cancer progression via increased proliferation, invasiveness, stemness, and chemoresistance. The downstream effectors, which have been shown to produce these effects on cell phenotype, are varied and often differ between cancer types (Table 2). HIF-1α is one of the main consistent mediators of the effects of P4HA1 overexpression, acting as both a transcription factor for P4HA1 and a downstream target affected by C-P4H activity [35,38,39,45]. Other downstream signaling pathways have been identified that contribute to proliferation, invasiveness, and stemness via regulation by P4HA1 overexpression. Downstream Wnt/β-catenin signaling in COAD is associated with increased proliferation and stemness [40,53]. There is also an association between P4HA1 expression and mitochondrial ferroptosis markers in COAD [84]. In nasopharyngeal carcinoma, P4HA1 overexpression is associated with upregulation of 3-hydroxy-3-methylglutaryl-CoA synthase 1 (HMGCS1), a component of the mevalonate pathway that contributes to ferroptosis resistance [85]. A major ECM remodeling protein, matrix metalloprotease 1 (MMP1), is upregulated by P4HA1 in PRAD, promoting invasion [44]. In GBM, upregulation of glycolytic metabolism and resultant effects on tumor growth by stabilization of phosphoglycerate kinase 1 (PGK1) [39] and increased invasiveness driven by increased CD31 display and VEGFA expression [86] are promoted by P4HA1 expression. Strikingly, these downstream signaling pathways that have been identified do not always correlate with changes in expression that have been identified using RNA-sequencing approaches, which may suggest that P4HA1 overexpression primarily exerts its effects at the protein level, as is the case with HIF-1α and PGK1.

**Table 2 ijms-26-09371-t002:** Effects of high P4HA1 expression in various cancer types.

Cancer Type	Downstream Effectors	Function(s)	References
Colorectal Cancer	Collagen I and IL17RB/c-Jun	Associated with increased metastasis in vitro and in vivo.	[70]
	HIF-1α	Associated with increased proliferation and stemness.	[40]
	Wnt/β-catenin (Canonical Wnt pathway)	Increases proliferation and stemness while reducing susceptibility to cell death.	[40,53]
Head and Neck Cancer	Collagen I	Increased collagen deposition is associated with altered cell cycle dynamics and increased cell stemness.	[68]
	HMGCS1	Increased proliferation and resistance to ferroptosis.	[85]
Breast Cancer	Collagen I	Increased deposition is associated with increased invasiveness via alteration of the ECM.	[51]
	HIF-1α	Positive feedback loop resulting in increased stemness and resistance to therapy.	[35]
Glioma	HIF-1α	Positive feedback loop associated with increased succinate production, altered glycolytic metabolism, and chemoresistance.	[39]
	PGK1	Succinylated via increased succinate production, increasing its stability and leading to altered glycolysis and lactate secretion.	[39]
	CD31 and VEGFA	Increased expression contributing to cell stemness, phenotypic plasticity, and shift towards endothelioid phenotype.	[86]
	YAP/Collagen I	Increase in hydroxylation of YAP, stabilizing it and leading to an increase in collagen I transcription and deposition. Increased collagen I deposition is associated with chemoresistance.	[87]
Pancreatic Cancer	HIF-1α	Positive feedback loop resulting in increased cell proliferation, resistance to chemotherapy, and stemness.	[45]
Prostate Cancer	MMP1	Increased invasion in vitro.	[44]
	Collagen I/DDR1/STAT3	Increased collagen secretion leads to autocrine activation of DDR1, activating downstream STAT3 to enhance cell invasion and stemness.	[67]

Overexpression of P4HA1 is frequently associated with increased collagen deposition, particularly of type I collagen. P4HA1 expression is associated with increased secretion of type I collagen in triple-negative breast cancer (TNBC) [35], COAD [70], PRAD [67], OVCAR [83], head and neck squamous carcinoma (HNSCC) [68], and melanoma [88]. Secretion of other collagen types can also be increased, such as type IV in GBM [36], melanoma [88], and uveal melanoma [38], type VI in GBM [86], and type X in chondrosarcoma [89]. Overexpression of P4HA1 may contribute to altered collagen fibril organization as well, such as in OVCAR, where knockdown of P4HA1 results in a shift in collagen fibril organization towards that which is seen in normal ovarian tissues [83]. Changes to collagen secretion are not cosmetic and have impacts on the cancer cell phenotype by modulating cell invasiveness, stemness, and proliferation. In prostate cancer, increased deposition of type I collagen allows cells to self-activate discoidin domain receptor tyrosine kinase 1 (DDR1), leading to activation of the STAT3 pathway to promote cell invasion and stemness [67]. Hypoxia in breast cancer can promote expression of P4HA1 via ATF3, which then increases collagen I deposition, remodeling the TME to favor invasion [51]. In HNSCC, lactate-driven secretion of type I collagen by HNSCC cells is associated with increased cell proliferation and stemness, as the phenotype produced by knockdown of COL1A1 expression can be rescued by addition of exogenous collagen [68]. As the primary known role of P4HA1 and C-P4H is to promote collagen secretion, these varied effects of altered collagen secretion and organization may provide insight into the selective pressures that result in the upregulation of P4HA1 in most solid cancers. In CRC, overexpression of P4HA1 driven by increased intracellular CHDH levels results in increased collagen I secretion that is associated with cell invasion both in vitro and in vivo [70]. Collagen secretion via P4HA1 may also be altered in response to extracellular factors, such as in GBM where treatment of cells with the drug temozolomide increased P4HA1 expression, increasing the rate of hydroxylation of yes-associated protein (YAP), which then drove expression of COL1 to promote chemoresistance [87]. While type I collagen secretion is increased in many cancer types, it has only been identified as having mechanistic impacts in a few cancer types; therefore, there is a need for studies on the role of both type I collagen and other collagens in promoting cancer progression due to P4HA1 overexpression.

P4HA1 has been associated with the hypoxic response and HIF-1α activity in many of the cancers in which it is overexpressed, including COAD [40], HSNCC [68], TNBC [35], GBM [39], PRAD [44], and pancreatic ductal adenocarcinoma (PDAC) [45]. The modulation of α-KG and succinate levels by increased C-P4H activity upon P4HA1 overexpression results in changes to the activity of HIF-associated prolyl hydroxylases that regulate HIF-1α stability, producing a positive feedback loop between P4HA1 and HIF-1α as expression of the former can be induced by the latter [35,39,45]. HIF-1α activity contributes to changes that promote cancer progression, such as increased stemness [35], proliferation [39,45], invasiveness [44,45], and chemotherapeutic resistance [35,45]. HIF-1α activity has also been indirectly related to increased collagen secretion with P4HA1 and/or C-P4H as a mediator, such as increased type I collagen in TNBC [35], luminal breast cancer [51], and head and neck squamous cell carcinoma (HNSCC) [68], and increased type VI collagen in uveal melanoma [38]. Knockdown of HIF-1α recapitulates a similar phenotype to P4HA1 knockdown while simultaneously reducing P4HA1 expression [35,39,45], further demonstrating that not only does P4HA1 increase HIF-1α activity, but also HIF-1α itself promotes P4HA1 expression. Altogether, this indicates that HIF-1α is both a central regulator and downstream effector of P4HA1 in cancer, implicating the hypoxic response as a major mechanism by which P4HA1 promotes cancer progression.

Overexpression of P4HA1 is also frequently associated with alterations to glucose metabolism in cancer. In many cases, these alterations have been linked to the hypoxic response, such as in PDAC, where crosstalk between P4HA1 and HIF-1α is associated with increased glucose uptake and lactate production [45,90]. Although there have not yet been molecular studies on identifying the mechanism(s), there is also evidence of a link between P4HA1, hypoxic response, and glycolytic changes in colon cancer [82], HNSCC [91], and osteosarcoma [92] using sequencing data from patients and publicly available databases. The mechanism is likely via the utilization of α-KG and production of succinate by the reaction catalyzed by C-P4H, driving glucose uptake via promoting entry of pyruvate into the citric acid cycle, depleting pyruvate levels, and driving glycolysis. Supporting evidence in GBM has demonstrated that increased levels of succinate due to P4HA1 overexpression can result in elevated PGK1 [39]. This increase in succinylated PGK1 reduces its rate of degradation and increases PGK1 protein level, promoting aerobic glycolysis as PGK1 is involved in the first ATP-generating step of glycolysis [39]. Furthermore, blocking uptake of glucose [90], lactate [67,68], or the conversion of lactate to pyruvate via LDHB [68] results in decreased collagen deposition, cell stemness, and proliferation, implicating pyruvate as the primary intermediate responsible. These mechanisms allow overexpression of P4HA1 to modulate cancer cells’ glycolytic metabolism, increasing rates of both glycolysis and the citric acid cycle to maintain C-P4H activity from P4HA1 overexpression.

## 5. P4HA1 Contributes to Cancer Progression by Altering the Cellular Immune Response and Promoting Remodeling of the ECM

RNA-seq data frequently identifies changes in tumor-infiltrating lymphocyte (TIL) abundance that correlates with P4HA1 expression, demonstrated in GBM [39], lung [78,93,94], colorectal [81,95], and head and neck cancer [96,97]. The magnitude of and cell types affected by these changes may be dependent on each individual cancer type; however, populations of CD8+ T cells, NK cells, monocytes, and macrophages are often affected. These changes also correspond to functional changes in TILs, particularly cytotoxic T cells [94,98] and microglia [99]. Using data from TCGA and GEO, high P4HA1 expression in COAD has been associated with alterations to immune cell infiltration, enrichment of cancer-associated fibroblasts (CAFs), and a possible negative correlation with response to PD-L1 checkpoint blockade therapy [84]. A combination of chloroquine to inhibit autophagy along with 5-fluorouracil to induce cytotoxicity increased the responsiveness of CD4+ and CD8+ T cells to CRC tissue while suppressing expression of several genes, including P4HA1, suggesting a role of P4HA1 in CRC immune evasion [98]. In a similar vein, indirect inhibition of P4HA1 in CAFs via semaglutide in a mouse model of PDAC with metabolic syndrome resulted in decreased collagen fibril deposition that allowed for an increase in CD8+ T cell infiltration knockdown of P4HA1 in a mouse model of GBM resulted in decreases in TIL abundance as well as IFN-γ and granzyme B levels, suggesting a functional reduction in cytotoxic cells [39]. Similarly, knockdown of P4HA1 in GBM resulted in polarization of microglia in the TME towards the anti-tumorigenic M1 phenotype and away from the pro-tumorigenic M2 phenotype while decreasing microglia motility towards GBM cells [99]. In CD8+ T cells, P4HA1 may aberrantly accumulate in the mitochondria under hypoxic conditions, leading to increased T-cell exhaustion and reducing anti-tumor immunity [7]. Here, knockout of P4HA1 or chemical inhibition of C-P4H using 1,4-dihydrophenonthrolin-4-one-3-carboxylic acid (1,4-DPCA) was able to increase the efficacy of CAR-T cells in vivo and increase the response to neoadjuvant therapy involving simultaneous PD-1 blockade to suppress tumor relapse and metastasis [7]. Altogether, these studies demonstrate that P4HA1 is an attractive therapeutic target not only for suppressing cell-intrinsic cancer progression, but also for improving immune response to cancer both innately and in response to immunotherapies.

Several studies have also shown that P4HA1 expression interacts with the TME in distinct ways, particularly with CAFs, and may even be upregulated in CAFs themselves. Lactate secreted by CAFs can be taken up by prostate cancer cells, which in turn promotes both P4HA1 expression and C-P4H activity via conversion to pyruvate and entrance into the citric acid cycle [67]. Factors secreted by the TNBC cell line MDA-MB-231 were able to promote a premetastatic niche in the lungs by influencing gene expression in lung fibroblasts, upregulating ECM remodeling proteins and downregulating ECM processing proteins, including P4HA1, influencing the distant ECM to be easier to invade through [100]. Other studies have shown P4HA1 to be overexpressed in this cell line and in TNBC in general [35,79,101], suggesting that P4HA1 may have various roles depending on where in the tumor or TME it is being expressed. In PDAC, both a mouse model of metabolic syndrome and PDAC patients with metabolic syndrome showed increased P4HA1 expression in CAFs, which was also associated with increased collagen deposition and disease aggressiveness, along with decreased T-cell infiltration [102]. These findings have demonstrated that expression of P4HA1 in both cancer cells and surrounding CAFs is important for producing a cancer-permissive TME.

## 6. Prospects and Future Directions

P4HA1 is overexpressed in a wide variety of cancers, ranging from TNBC and COAD to PDAC. Bioinformatic and in vitro studies have demonstrated that overexpression of P4HA1 contributes to cancer progression in multiple ways, including proliferation, metastasis, and resistance to therapy. While these studies have reported trends in patient data and/or insights into the function of P4HA1 using in vitro functional assays, there remains a need to characterize the molecular mechanisms that drive the oncogenic effects of P4HA1 overexpression. Fewer studies have reported downstream signaling pathways associated with P4HA1 overexpression than have reported upstream regulators of P4HA1 expression and activity. As P4HA1 overexpression contributes to phenotypic changes other than just those which directly relate to collagen deposition and ECM remodeling, mechanistic investigation of P4HA1 should also include potential alternative substrates for hydroxylation. Furthermore, there are few studies in vitro that aim to assess the efficacy of P4HA1 as a therapeutic target and even fewer in vivo, demonstrating a clear need for further studies using P4HA1-targeted therapies. Encouragingly, a few recent studies have reported promising results using systems other than in vitro approaches, such as using P4HA1-knockout mice to study the regulation of P4HA1 in vivo [87,103] and an ex vivo patient-derived organoid model of PDAC demonstrating a role of P4HA1 in hypoxia-associated invasion [104]. Continuing to use these models to uncover the molecular mechanisms of P4HA1 and evaluate its potential as a therapeutic target will be essential to better our understanding of such a commonly upregulated gene in cancer.

C-P4H may have substrates other than collagen, resulting in changes to hydroxylation upon P4HA1 overexpression that may impact pathways other than those directly influenced by collagen and its related signaling pathways. In melanoma, secretion of the protein collagen triple helix repeat containing 1 (CTHRC1) is a negative prognostic marker and is thought to be a mediator of melanoma migration and invasion [88]. Secretion of this protein, in particular a large molecular weight, possibly trimerized, isoform, is significantly reduced when P4HA1 is knocked out or inhibited with 3,4-dihydroxybenzoic acid (3,4-DHB) [88]. As CTHRC1 has a short collagen motif with 12 -X-Y-Gly repeats [105], it is possible that hydroxylation of these collagen repeats by C-P4H may occur to facilitate the formation of dimers and trimers, contributing to their secretion and facilitating a metastatic phenotype in melanoma. Mannose-binding lectins (MBLs) undergo prolyl hydroxylation and other post-translational modifications on several collagen-like domains to form active oligomers, which are important for innate immunity [106]. Recently, C-P4H was identified as the dioxygenase responsible for this proline hydroxylation, indicating that it may serve roles in hydroxylating non-collagen molecules in normal function as well as in cancer [106]. YAP1, a transcriptional coactivator in the Hippo pathway that has been associated with cell proliferation and invasiveness in multiple cancer types, can be hydroxylated by C-P4H, preventing its degradation and increasing expression of its downstream targets, which include COL1 [87,107]. YAP1 is not known to have a collagen-related domain, indicating that C-P4H may be more promiscuous than recognizing only collagen and collagen-like sequences. Although the known role of C-P4H is to facilitate the folding and secretion of collagen, its ability to hydroxylate other substrates may contribute to the wide variety of signaling pathways associated with P4HA1 overexpression. This is a promising area of research for improving our understanding of C-P4H and the potential for targeting it and/or P4HA1 with therapeutic approaches.

In vitro studies have demonstrated the efficacy of using C-P4H inhibitors such as 1,4-DPCA [35,45,108], 3,4-DHB [24,45,67], and diethyl pythiDC [24,59]; however, few studies have done so in vivo. Those that have been published reported encouraging data, increasing susceptibility to docetaxel in TNBC [35], reducing lung metastasis from COAD xenograft tumors [70], and decreasing tumor growth in a PDX mouse model of CRC [62]. Additionally, these in vivo studies have demonstrated that these treatments are safe in mice despite using small-molecule inhibitors that are not specifically targeted to cancer cells. There still remains concern for unintended side effects due to the ubiquity of C-P4H and collagen throughout the body, which may contribute to difficult dosing in clinical settings. Some small molecule inhibitors of C-P4H also present with potential off-target effects, which complicate clinical usage, particularly as a monotherapy; for example, 3,4-DHB may cause iron deficiency at effective concentrations due to its affinity for iron [24]. Diethyl pythiDC is a similar inhibitor that shows negligible affinity for iron and may be more suitable as a therapy in that regard [24]; however, its safety has not been evaluated as extensively as 3,4-DHB or 1,4-DPCA. Another option that may reduce off-target effects is the usage of siRNA to specifically silence P4HA1, paired with a targeted delivery system. One study thus far has used chitosan–gelatin particles containing P4HA1 siRNA, which were effective against glioma cells both in vitro and in vivo without significant toxicity [109]. The strength of various bioinformatic studies, along with these studies in vitro and in vivo, has demonstrated promise in expanding studies into P4HA1 in cancer, particularly for those cancers in which current research is encouraging but lacking, such as prostate cancer, ovarian adenocarcinoma, and melanoma. Furthermore, the utility of targeting P4HA1 as a combination treatment should also be evaluated, particularly as its overexpression has frequently been associated with increased cancer cell stemness, a major contributor to treatment resistance. Combining P4HA1 inhibition or silencing with a cytotoxic treatment may therefore be more effective than either treatment alone.

## 7. Conclusions

In summary, here we have presented a comprehensive narrative review of the current literature on C-P4H structure and function and the role of its α subunit, P4HA1, in promoting cancer progression. The current literature has been described with brevity in order to cover the wide variety of functions via which P4HA1 contributes to cancer progression and provides an overview of current efforts to target it therapeutically. P4HA1 expression and/or activity are activated by many different types of signals and transcription factors (Figure 2). Overexpression of P4HA1 is seen in a wide variety of solid cancer types and is frequently associated with poorer overall survival, which has been implicated in increasing cancer proliferation, invasion, stemness, therapy resistance, ECM remodeling, and angiogenesis, as well as priming of TILs in the TME to a pro-tumorigenic state (Figure 2). Therefore, P4HA1 is a promising prognostic marker and therapeutic target in a wide variety of solid cancers.

## Figures and Tables

**Figure 1 ijms-26-09371-f001:**
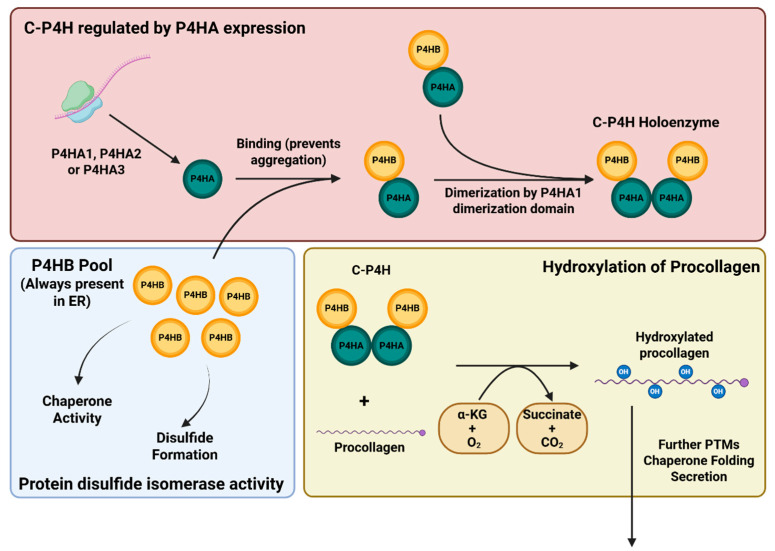
Schematic of C-P4H formation from P4HA and P4HB and hydroxylation of procollagen by C-P4H. (Created in BioRender. Hironaka, D. (2025) https://BioRender.com/jpge1gu, accessed on 21 September 2025).

**Figure 2 ijms-26-09371-f002:**
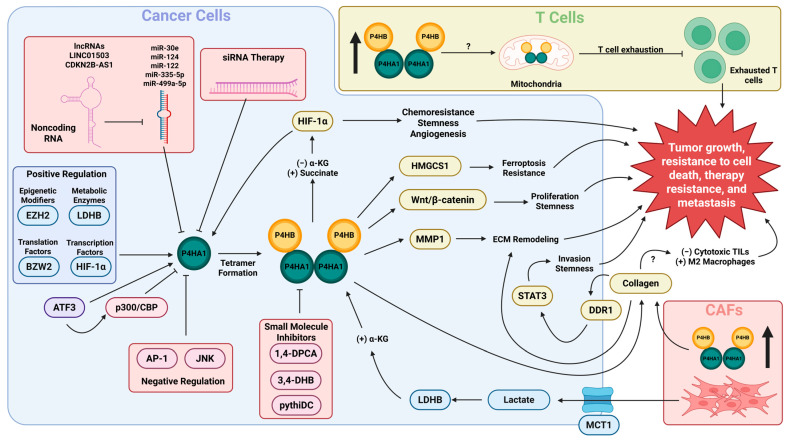
Schematic overview of C-P4H regulatory mechanisms, downstream effectors, and contributions to cancer progression. Arrowheads indicate increased expression or activation. Flatheads indicate inhibition of expression or activity. Question marks (?) above arrows indicate that the resulting phenotype has been demonstrated empirically but the mechanism has not yet been described. (Created in BioRender. Hironaka, D. (2025) https://BioRender.com/f26z331, accessed on 21 September 2025).

## Data Availability

No new data were created or analyzed in this study. Data sharing is not applicable to this article.

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
