# Peer review of "Enhanced Collagen Prolyl 4-Hydroxylase Activity and Expression Promote Cancer Progression via Both Canonical and Non-Canonical Mechanisms"

_ijms, 2025, doi:10.3390/ijms26199371_

Round 1

Reviewer 1 Report

Comments and Suggestions for Authors

see attachment.

Author Response

We appreciate your positive feedback and think that “our topic is relevant to the field” “study design is well defined”.  Here is our reply to your comments:

Comments 1: The references are appropriate for the role of C-P4H in tumor progression and include recent papers. However, some other papers on the topic have been published in the meantime (e.g., papillary thyroid cancer; pancreatic cancer) and could further update the data of this review.

Response 1: We agree with this comment and have updated the data in our review accordingly. Specifically, we have included several additional bibliographic references in the revised manuscript (see page 6, line 246-252; page 8, line 291-293; page 10, line 386-389; page 12, line 441-444), which are highlighted in red.

Comments 2: In Table 1 the tumor type should be added in correspondence with the bibliographic references related to miR-30, miR-122 3 miR-124. 

Response 2: Thank you for pointing this out. We have added the tumor types for the bibliographic references related to miR-30, miR-122 and miR-124 in Table 1 (see revised Table 1 in page 7).

Reviewer 2 Report

Comments and Suggestions for Authors

The manuscript “Enhanced collagen prolyl 4-hydroxylase activity and expression promote cancer progression via both canonical and non-canonical mechanisms” by Dalton Hironka and Gaofeng Xiong addresses a focused and timely topic and is generally well written. The authors demonstrate thorough knowledge of the field, provide informative figures and tables, and give a balanced discussion of canonical and non-canonical roles of C-P4H and P4HA1 in tumor biology.

The principal shortcoming is methodological opacity - the paper is presented as a literature review but does not describe the article selection process. The authors should clarify the review aim (e.g., narrative synthesis, scoping review, or systematic review) and, if any systematic elements were used, report databases searched, search terms or syntax, date ranges, inclusion/exclusion criteria, and screening/selection procedures (or explicitly state that an expert narrative selection was applied and justify the principles guiding it). Absence of these details limits reproducibility and prevents assessment of coverage bias.

Content coverage appears broad and up-to-date, yet without a documented search strategy it is impossible to judge completeness or potential omission of conflicting or negative evidence. I therefore recommend adding either a brief methods subsection describing search and selection, or a short explicit statement explaining an expert/narrative approach. A concise paragraph in the Discussion should also summarize dominant study types (in vitro, in vivo, clinical, bioinformatics) and identify clear gaps in the literature.

Technically, the manuscript is readable and figures/tables are useful. After the methodological clarification and a short limitations paragraph, the paper will be suitable for publication. 

Author Response

We appreciate your positive feedback and consider our manuscript is “generally well written” and “suitable for publication” after minor revision. Here is our reply to your comments:

Comments 1: The principal shortcoming is methodological opacity - the paper is presented as a literature review but does not describe the article selection process. The authors should clarify the review aim (e.g., narrative synthesis, scoping review, or systematic review) and, if any systematic elements were used, report databases searched, search terms or syntax, date ranges, inclusion/exclusion criteria, and screening/selection procedures (or explicitly state that an expert narrative selection was applied and justify the principles guiding it). Absence of these details limits reproducibility and prevents assessment of coverage bias.

Content coverage appears broad and up-to-date, yet without a documented search strategy it is impossible to judge completeness or potential omission of conflicting or negative evidence. I therefore recommend adding either a brief methods subsection describing search and selection, or a short explicit statement explaining an expert/narrative approach.

Response 1: We appreciate the reviewer’s thoughtful comment regarding methodological transparency. We confirm that this manuscript is intended as a narrative review, aimed at synthesizing and contextualizing the most relevant and recent developments in the role of P4AH1 in cancer research area. In response to the reviewer’s suggestion, we have added a brief description in the revised manuscript (see page 13, line 499-503) to clarify this narrative approach and outline the principles guiding literature selection.

Comments 2: A concise paragraph in the Discussion should also summarize dominant study types (in vitro, in vivo, clinical, bioinformatics) and identify clear gaps in the literature.

Response 2: We thank the reviewer for this insightful comment. We have revised the concise paragraph in the discussion (Prospects and future directions) to summarize dominant study types (see page 11, line 428-page 12, line 447).